# Developing random forest hybridization models for estimating the axial bearing capacity of pile

**Tuan Anh Pham, Van Quan Tran** *

University of Transport Technology, Hanoi, Vietnam

* quantv@utt.edu.vn

## Abstract

Accurate determination of the axial load capacity of the pile is of utmost importance when designing the pile foundation. However, the methods of determining the axial load capacity of the pile in the field are often costly and time-consuming. Therefore, the purpose of this study is to develop a hybrid machine-learning to predict the axial load capacity of the pile. In particular, two powerful optimization algorithms named Herd Optimization (PSO) and Genetic Algorithm (GA) were used to evolve the Random Forest (RF) model architecture. For the research, the data set including 472 results of pile load tests in Ha Nam province—Vietnam was used to build and test the machine-learning models. The data set was divided into training and testing parts with ratio of 80% and 20%, respectively. Various performance indicators, namely absolute mean error (MAE), mean square root error (RMSE), and coefficient of determination ($R^2$) are used to evaluate the performance of RF models. The results showed that, between the two optimization algorithms, GA gave superior performance compared to PSO in finding the best RF model architecture. In addition, the RF-GA model is also compared with the default RF model, the results show that the RF-GA model gives the best performance, with the balance on training and testing set, meaning avoiding the phenomenon of overfitting. The results of the study suggest a potential direction in the development of machine learning models in engineering in general and geotechnical engineering in particular.

## 1. Introduction

In engineering, piles have been used to support the building, in which, the axial load capacity of the pile is considered the most important parameter in the pile foundation design. Typically, the axial load capacity of a pile can be determined using static and dynamic load tests on the construction sites. However, these methods are not only time-consuming and expensive but also often difficult to apply to small-scale projects [1, 2]. Therefore, several other approaches have been proposed in predicting the axial load capacity of piles and improving the prediction accuracy. These methods include the use of empirical approaches based on in-situ test results, such as SPT (Standard Penetration Test), CPT (Cone Penetration Test), and geometrical

**Data Availability Statement:** All relevant data are within the paper and its Supporting Information files.

**Funding:** The authors received no specific funding for this work.

**Competing interests:** The authors have declared that no competing interests exist.

parameters of the pile [3–8]. The empirical formulas include a few key parameters, so it is not enough to accurately predict the pile load capacity [8]. In addition, the use of several experimental coefficients, which have a wide range for different types of soil, further deviates from the actual results [9].

Over the past two decades, artificial intelligence (AI) and machine learning (ML) have made great progress, being applied to solve many real-world problems in general and engineering in particular. In details, some ML techniques have been used in solving many engineering problems such as geotechnical problems [10–13], mechanism properties of materials [14–20], rock blasting [21]. In details, Armaghani et al. [13] developed a hybrid ML model including artificial neural network (ANN) and particle swarm optimization (PSO) in predicting settlement of pile. Using hybrid Ensembling of Surrogate ML models, Asteris et al. [14, 18, 22] improved the accuracy of ML model in predicting compressive strength of concrete. Apostolopoulou et al. [16] develop ANN model in designing natural hydraulic lime mortars. Using an ANN model, Armaghani et al. [17] can predict the unconfined compressive strength of granite with only two input variables. Armaghani et Asteris [19] propose ANN and ANFIS models in predicting the compressive strength of cement based mortar with high performance. Developing hybrid ML model including meta-heuristic search of sociopolitical algorithm and Extreme Gradient Boosting to predict compressive strength of recycled aggregate concrete. The axial load capacity of concrete filled steel tube columns can be estimated by ANN model in the investigation of Le et al. [15]. In rock blasting, the peak particle velocity can be successfully predicted by support vector machine (SVM) [21] which is famous ML technique.

In this clear trend, many studies apply artificial intelligence to solve the problem of estimating the bearing capacity of piles. For example, Kumar [23] developed a k-nearest neighbor (KNN) model to predict the soil parameters required for foundation design. Goh [24, 25] presented an ANN model to predict the bearing capacity of driven piles in clays. Besides, Shahin [26] developed an ANN model to estimate the bearing capacity of driven piles and drilled shafts using a series of in-situ load tests. Nawari [27] showed an ANN algorithm to predict the deflection of drilled shafts based on (SPT) data and the shaft geometry. Momeni [28] developed ANN models to predict the shaft and tip resistance of concrete piles. Pham et al. [10] presented two models, including ANN and RF to estimate the ultimate bearing capacity of the driven pile. Shahin and Jaksa [29] presented an ANN model to predict the bearing capacity of the drilled shaft using CPT data.

The published literatures show that AI has good potential to accurately predict the load capacity of piles. However, it must be said that ML models have a very complex architecture, including many hyperparameters. These hyperparameters are particularly sensitive and greatly affect the model's forecast results [11, 30–32]. The above studies did not show that how the model architecture model is selected to predict the pile load capacity. The choice of model architecture is usually done manually, which takes a lot of time and resources. As mentioned above, various studies have been carried out to evaluate the performance of ML algorithms in predicting pile bearing capacity. However, creating hybrid models using optimization algorithms to choose the best model is a matter of concern. RF model has been proving to be one of the best ML algorithms, achieving excellent performance in previous studies [10, 33, 34]. As a matter of course, there are many optimization algorithms used to solve the problem in techniques such as gradient descent [35], quasi-newton [36], hill climb [37], simulated annealing [38], particle swarm optimization (PSO) [39], and genetic algorithm (GA) [40]. Among those algorithms, GA and PSO do not use problem gradients to be optimized, it does not require optimization problems to be as distinct as standard optimization methods such as gradient descent and quasi-newton [41]. Therefore, these are two of the most powerful and popular algorithms today in solving general engineering problems.

From the above analysis, the main objective of this present investigation is to apply the two-hybrid soft computing model RF-GA and RF-PSO for the better and quick prediction of axial bearing capacity of piles based on the 10 parameters of piles geometry and soil properties. To acquire this aim, a database consisting of 472 pile load tests collected from the available literature [11]. Various performance criteria including the coefficient of determination (R2), root mean squared error (RMSE), and the mean squared error (MAE) are considered to evaluate the prediction capability of the two-hybrid RF models and individual model RF. Furthermore, 1000 simulations taking into account the randomness of the model inputs were performed to fully evaluate the feasibility of these models.

## 2. Research significance

High performance estimation of axial bearing capacity of pile is meaningful due to foundation design and contributions to building design. Although some machine learning models were developed to predict the axial bearing capacity of pile. For instance, ANN model in the investigation of Shahin et al. [26, 29], however low number of data containing 80 samples, that were used to develop the ANN model, reduces the performance and reliability of ML model in predicting the axial bearing capacity of pile. The Random Forest model developed by Pham et al. [10] has performance values of prediction as following $R^2 = 0.866$, RMSE = 0.0982 MN, MAE = 0.2924 MN. This performance of RF model can be improved. Thus, the following might emphasize various contributions of the current investigation: 0.9331, 0.0929, 0.0675

1. A database containing 10 input variables and 472 samples is presented;

2. The hybrid models RF-GA and RF-PSO are developed to find the best hyperparameters of Random Forest model for predicting the axial bearing capacity of pile;

3. Monte Carlo simulations are introduced to evaluate the performance and reliability of single RF, RF-GA and RF-PSO;

4. The performance of axial bearing capacity prediction is increased by using hybrid model RF-GA;

5. A sensitivity analysis is performed with aided Shapley Additive Explanations to reveal the effects of input variables on both magnitude of axial bearing capacity of pile and performance prediction of RF-GA model.

## 3. Database construction

The data used for this study were obtained from published literature [11]. To correctly predict the bearing capacity of piles, a thorough understanding of the factors that affect the bearing capacity of the pile is needed. Most traditional pile bearing capacity determination methods include the following parameters: pile geometry, pile material properties, and soil properties [3, 42, 43]. Since SPT is one of the most popular in-situ tests, the soil properties were characterized through SPT results. In this study, the average of SPT values along the pile shaft and pile tip is taken as the main input to determine the bearing capacity of the pile. In addition, information on pile geometry and thickness of soil layers are also collected to ensure sufficient factors are used for determining pile bearing capacity [3]. More specifically, the input parameters for the model include (i) Pile diameter ($X_1$); (ii) length of pile tip segment ($X_2$); (iii) length of 2nd pile segment ($X_3$); (iv) length of pile top segment ($X_4$); (v) the natural ground elevation ($X_5$); (vi) pile top elevation ($X_6$); (vii) guide pile segment stop driving elevation ($X_7$); (viii) pile tip elevation ($X_8$); (ix) the average SPT blow along the embedded length of the pile ($X_9$) and (x)

the average SPT blow at the tip of the pile ($X_{10}$). The diagram of pile parameters was shown in Fig 1. The bearing capacity is the single output variable in this study ($P_u$).

As observed in Table 1, the pile diameter ($X_1$) ranged from 0.3 to 0.4 m. The length of the pile tip section ($X_2$) ranged from 3.4 to 5.7 m. The length of the second pile segment ($X_3$) ranged from 1.5 to 8 m. The length of the pile top segment ($X_4$) ranged from 0 to 1.69 m which 0 value means that the segment does not exist. The natural ground elevation ($X_5$) varied from 0.68 to 3.4m. The pile top elevation ($X_6$) varied from 3.04 to 4.12 m. The guide piles' stop driving elevation ($X_7$) varied from 1.03 to 4.35m. The pile tip elevation ($X_8$) varied from 8.3 to

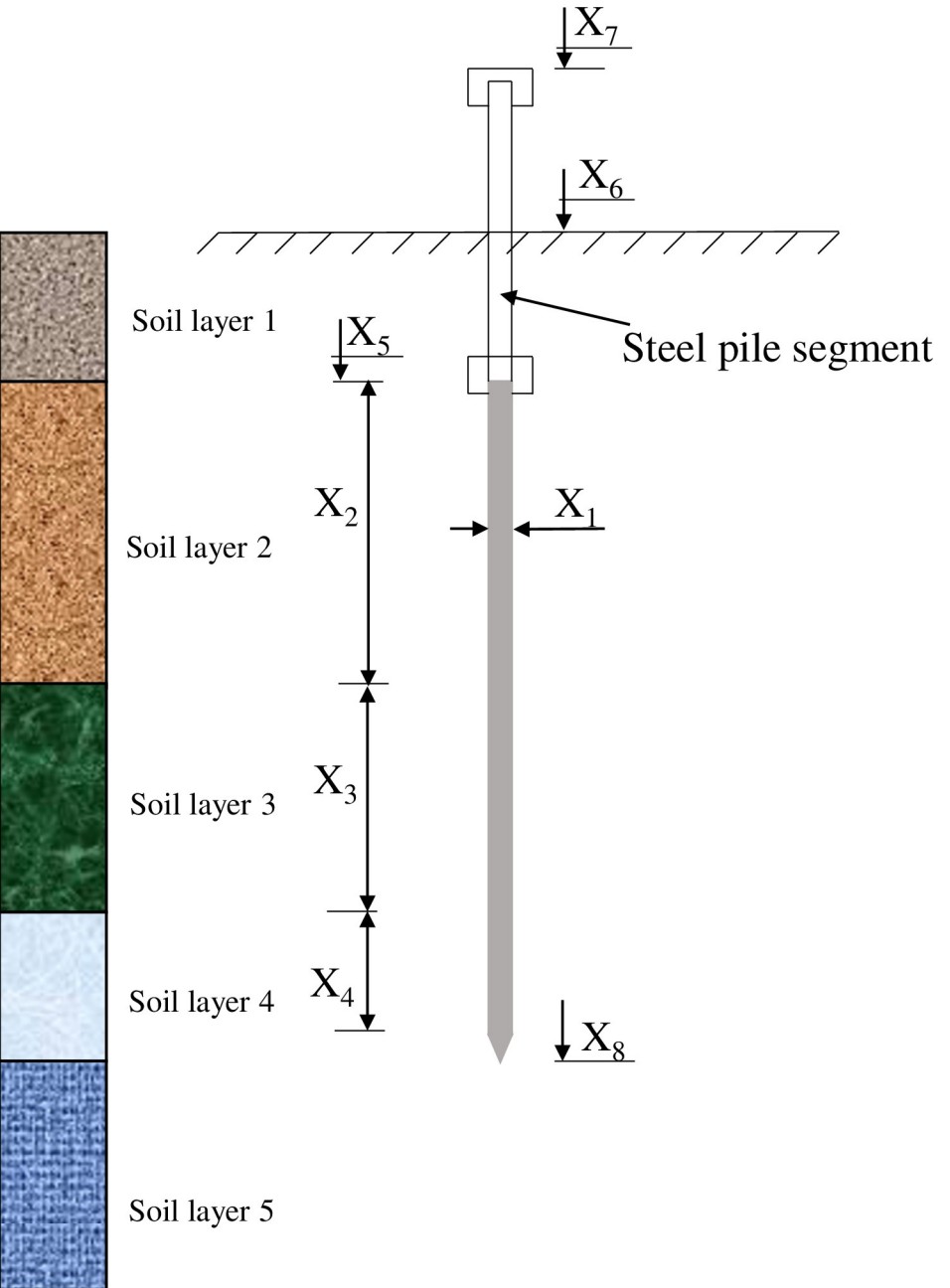

**Fig 1. Diagram of pile parameters.**

**Table 1. Inputs and output of the present study.**

|  | X₁ | X₂ | X₃ | X₄ | X₅ | X₆ | X₇ | X₈ | X₉ | X₁₀ | Pu |
|---|---|---|---|---|---|---|---|---|---|---|---|
| Unit | m | m | m | m | m | m | m | m | - | - | MN |
| Count | 472 | 472 | 472 | 472 | 472 | 472 | 472 | 472 | 472 | 472 | 472 |
| SD(*) | 0.048 | 0.482 | 1.638 | 0.457 | 0.616 | 0.080 | 0.599 | 1.798 | 2.264 | 0.660 | 0.353 |
| Min | 0.3 | 3.4 | 1.5 | 0 | 0.68 | 3.04 | 1.03 | 8.3 | 5.6 | 4.38 | 0.407 |
| Mean | 0.364 | 3.826 | 6.579 | 0.331 | 2.804 | 3.495 | 2.918 | 13.538 | 10.743 | 7.056 | 0.984 |
| Median | 0.4 | 3.45 | 7.31 | 0 | 2.95 | 3.48 | 3.275 | 14.11 | 10.8 | 7.175 | 1.069 |
| Max | 0.4 | 5.70 | 8 | 1.69 | 3.4 | 4.12 | 4.35 | 16.09 | 15.41 | 7.75 | 1.551 |

SD$^{(*)}$ = Standard deviation.

16.09 m. The average SPT blow along the embedded length of the pile ($X_9$) ranged from 5.6 to 15.41. The average SPT blow at the tip of the pile ($X_{10}$) ranged from 4.38 to 7.75. The bearing capacity load ($P_u$), ranged from 0.407 MN to 1.551 MN with a mean value of 0.984 MN and a standard deviation of 0.353 MN.

The data distribution between the input variables and axial bearing capacity is plotted in Fig 2, the linear correlation coefficients are shown in Fig 3. As Fig 1 clearly shows, some input variables are significantly correlated such as $X_3$ and $X_8$, $X_9$ and $X_8$. However, all input variables are considered in this investigation to increase the accuracy of the proposed model.

In this investigation, the collected dataset was divided into the training and testing datasets. The number of samples used for training should not be too small, so it will be difficult for the model to learn the generality of the data. In addition, because the number of samples is quite large, the selected sample ratio is 80% for training and 20% for testing in this study, still ensuring that the number of test samples is enough to confirm the model performance. Different from the original data, the training dataset (including 10 inputs and 1 output) was normalized in the [0; 1] range to help variables have the same importance. A normalization process of parameters, such as the minimum and maximum values of the training data were performed to scale the testing dataset.

## 4. Methods used

### 4.1. Random forest (RF)

Randomized Forest (RF) belongs to the family of ML methods, which includes different algorithms for generating a set of decision trees. The random forest method was first proposed by Ho [44], and quickly became one of the powerful ML algorithms, commonly used to solve various problems [10, 34, 45]. In essence, RF was a bagging ensemble method that can improve variable selection [46]. Breiman [47] showed that random forests which are grown using random vectors in the tree construction are equivalent to a kernel acting on the true margin. In this algorithm, two principles of "randomization" are used: Bagging and Random Feature Selection [48]. That is, each decision tree in a random forest was built based on a random number of input features. Therefore, the RF model adjusts the decision tree's over-fitting habits into their training set, or in other words, the RF generally outperforms the decision tree. A general randomized forest model is shown in Fig 4.

When Breiman introduced the RF model in [47], the author also demonstrated that when the number of trees exceeds a certain value, adding other trees does not systematically improve the performance of the RF. This result suggests that the number of trees in RF does not need to be too large to achieve a high-efficiency performance [45, 49].

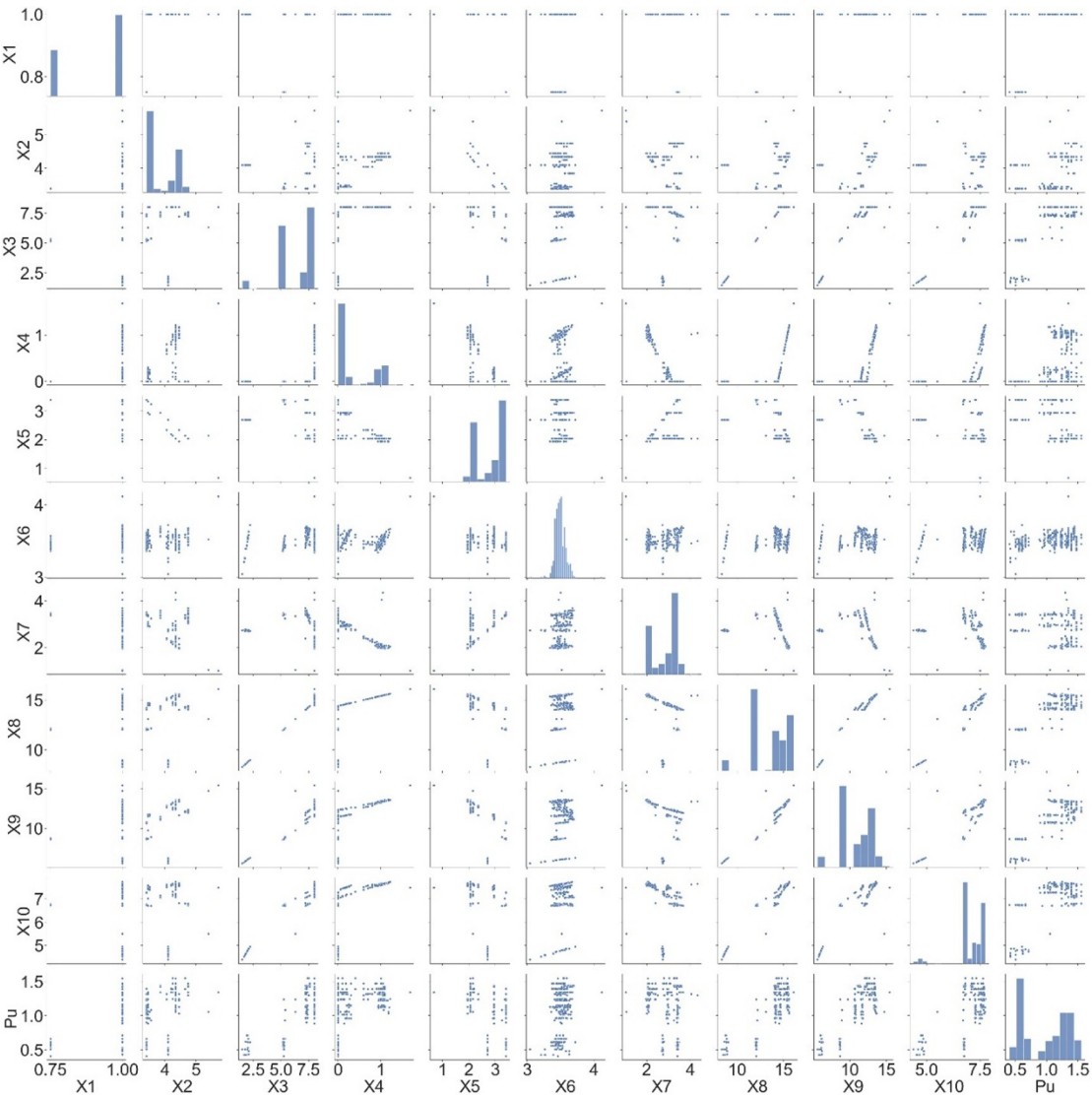

**Fig 2. Data distribution of input variables and output $P_u$.**

In this study, the RF model used from the scikit-learn library [50], and five model hyper-parameters that have the greatest influence on the predicted results of the RF model are considered. To be more specific, these include $H_1$—the maximum depth of the tree; H2—the maximum features used for random bagging in each decision tree; $H_3$—the minimum number of samples required to be at a leaf node; $H_4$—the minimum number of samples required to split a node; $H_5$ –the number of the decision tree.

While the number of trees does not need to be too large, parameters such as $H_1$, $H_2$, $H_3$, $H_4$ affect the complexity of the tree. Trees that are too complex can cause the model to over-fitting and not achieve high generalization.

## 4.2. Particle swarm optimization (PSO)

Particle Swarm Optimization (PSO) was one of the most widely used optimization techniques. J. Kennedy and R. Eberhart [39] were the first to present it. It became famous due to the fact

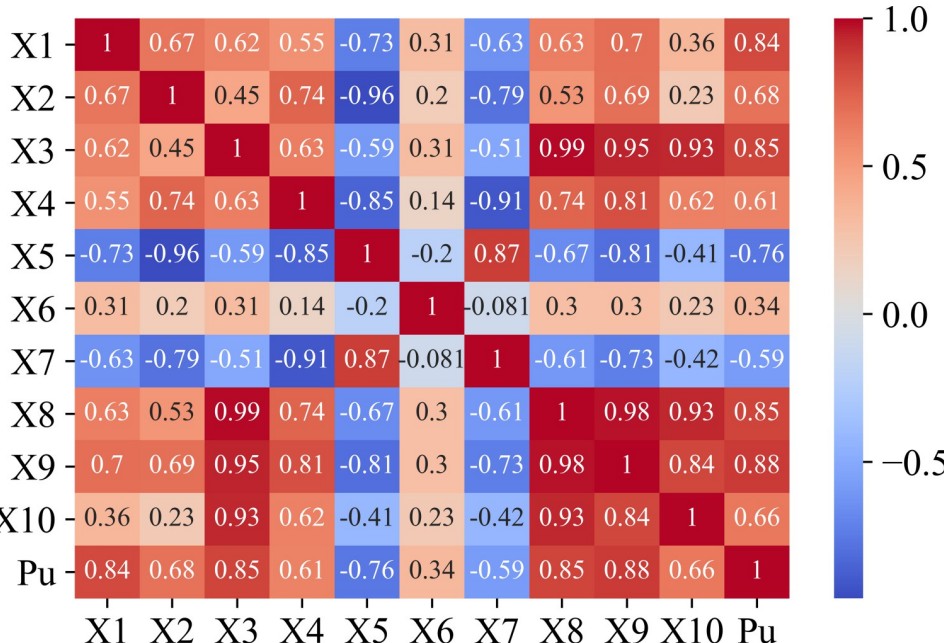

**Fig 3. Relation between input and output via matrix Pearson correlation.**

that it was a type of continuous optimization procedure. PSO employs shifting the position of the particles in the herd at a constant velocity that is updated with each iteration to find the optimal solution. Each particle's mobility is influenced by the swarm's personal best position and global best (cf. Fig 5). PSO is widely employed for optimization issues in various domains of engineering, particularly geotechnical [51, 52].

The pseudo-code of the algorithm is presented below:

```
FOR each particle i in swarm
  FOR each dimension j
    Initialize position Gij randomly
    Initialize velocity Vij randomly
  END FOR
END FOR
Iteration k = 1
DO
  FOR each particle i in swarm
    Calculate fitness value P(i)
      IF P(i) > P_best(i) THEN
        P_best(i) = P(i)
      END IF
      IF P(i) > G_best THEN
        G_best = P(i)
      END IF
  END FOR
  FOR each particle i in swarm
    FOR each dimension j
      Calculate new velocity:
      Vij(k+1) = wVij(k) + c1rand1(P_best(i)−Gij(i)) + c2rand1 (G_best−
      Gij(i))
      Update particle positon: Gij(k+1) = Gij(k) + Vij(k+1)
    END FOR
  END FOR
```

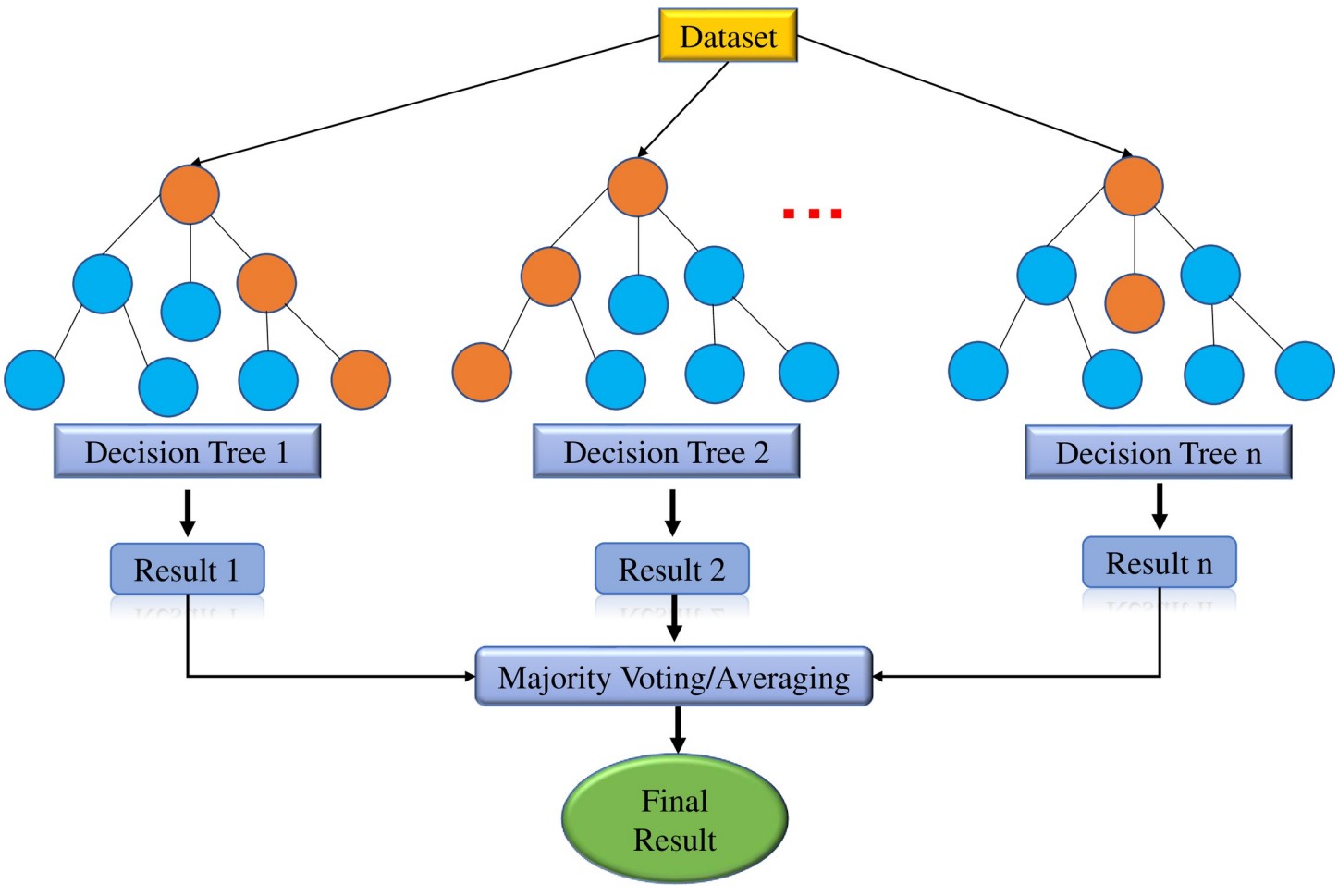

**Fig 4. Random forest model.**

```
w = w.w_d
k = k + 1
```
**WHILE k <** maximum_Iteration

In which: w is an inertial parameter; $c_1$, $c_2$ are the acceleration coefficients; $w_d$ is the reduction coefficient of w.

## 4.3. Genetic algorithm (GA)

GA is one of the most powerful global optimization algorithms, used to solve various problems. This Algorithm was first introduced by Holland [40]. The origins of this approach are based on the Darwinian theory, in which an evolved and adaptable population rests on the most powerful individuals. In the GA algorithm, the population size is one of the most important factors reflecting the total number of solutions and significantly affects the results of the problem [53], while the number of generations refers to the maximum number of iterations of the algorithm [54]. Same as PSO, GA does not use gradient descent, so GA allows finding the minimum of a function even in the absence of a derivative. Moreover, other studies used the GA method whose effectiveness has been proven [6, 53, 55–57].

In this study, using the GA algorithm, an optimization technique was developed to find the optimal model architecture for RF.

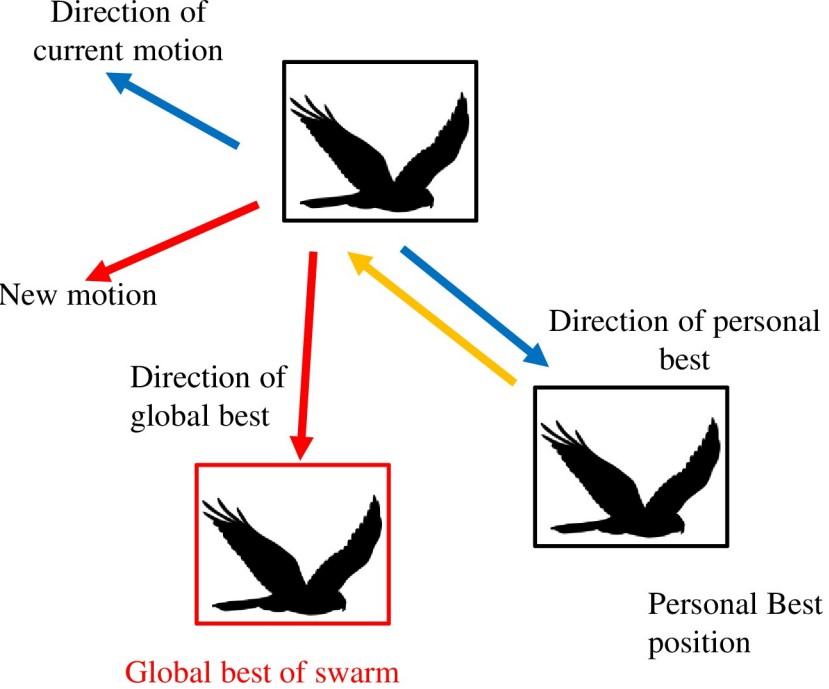

**Fig 5. Particle movement by the swarm direction.**

The pseudo-code of the GA algorithm is presented below:

```
FOR each chromosome i in Population
  FOR each gene j
    Initialize Gij randomly
  END FOR
END FOR
Generation k = 1
DO
  FOR each chromosome i in Population
    Calculate the fitness value Pi
  End FOR
    Mating the best chromosomes
    Mutates some children randomly
    Remove the weakest chromosomes
k = k + 1
WHILE maximum generation
```

## 4.4. Modeling and hyper-parameters tuning

In this investigation, the RF model is proposed in modeling the nonlinear relationship between the inputs and the output. To get high performance, the hyper-parameters of RF will be tuned using optimization algorithms including GA and PSO. Five parameters of the RF model are tuned as suggested in the literature [33]. Table 2 showed the tuned hyper-parameters, the explanation, and the value of tuning ranges. Thus, the architecture of the population in GA (or the swarm in PSO) was illustrated in Fig 6. It can be seen that the population (or the swarm) has many members, and each member has five dimensions corresponding to the 5 hyperparameters of the RF model. The flowchart of hybrid RF models was illustrated in Fig 7. In these models, RF was used as the fitness function and the member with the best fitness value was

**Table 2. Hyper-parameters description and tuning range.**

| No | Denote | Hyperparameters | Explanation | Range |
|---|---|---|---|---|
| 1 | $H_1$ | Max_depth | The maximum depth of decision tree | 2–20 |
| 2 | $H_2$ | Max_features | The maximum features which random chosen for bagging. | 1–10 |
| 3 | $H_3$ | Min_samples_leaf | The minimum number of samples required to be at a leaf node | 2–20 |
| 4 | $H_4$ | Min_samples_split | The minimum number of samples required to split an internal node | 2–20 |
| 5 | $H_5$ | n_estimators | The number of trees in the forest | 2–200 |

considered as the best individual. The dimension value of the best individual are selected as the best hyperparameters of the RF model.

For more objective results, 20 models RF-GA and RF-PSO were developed, taking into account the random initialization of the population. To make the comparison between the optimization algorithms, the maximum number of iterations of the two-optimization algorithms is 100 and the number of the population is 30. It is important to note that, to avoid overfitting of the models to the data, the 10-fold CV technique on the training set was used in this step. In this technique, the training data set is divided into 10 folds, 9 folds for training, and 1-fold for verification. The average results of 10 such times were compared for each optimization iteration step to confirm the performance of the hybrid models. All initial parameter setting in the GA and PSO was determined by trial tests [33]. The best initial parameter settings for GA and PSO were given in Table 3.

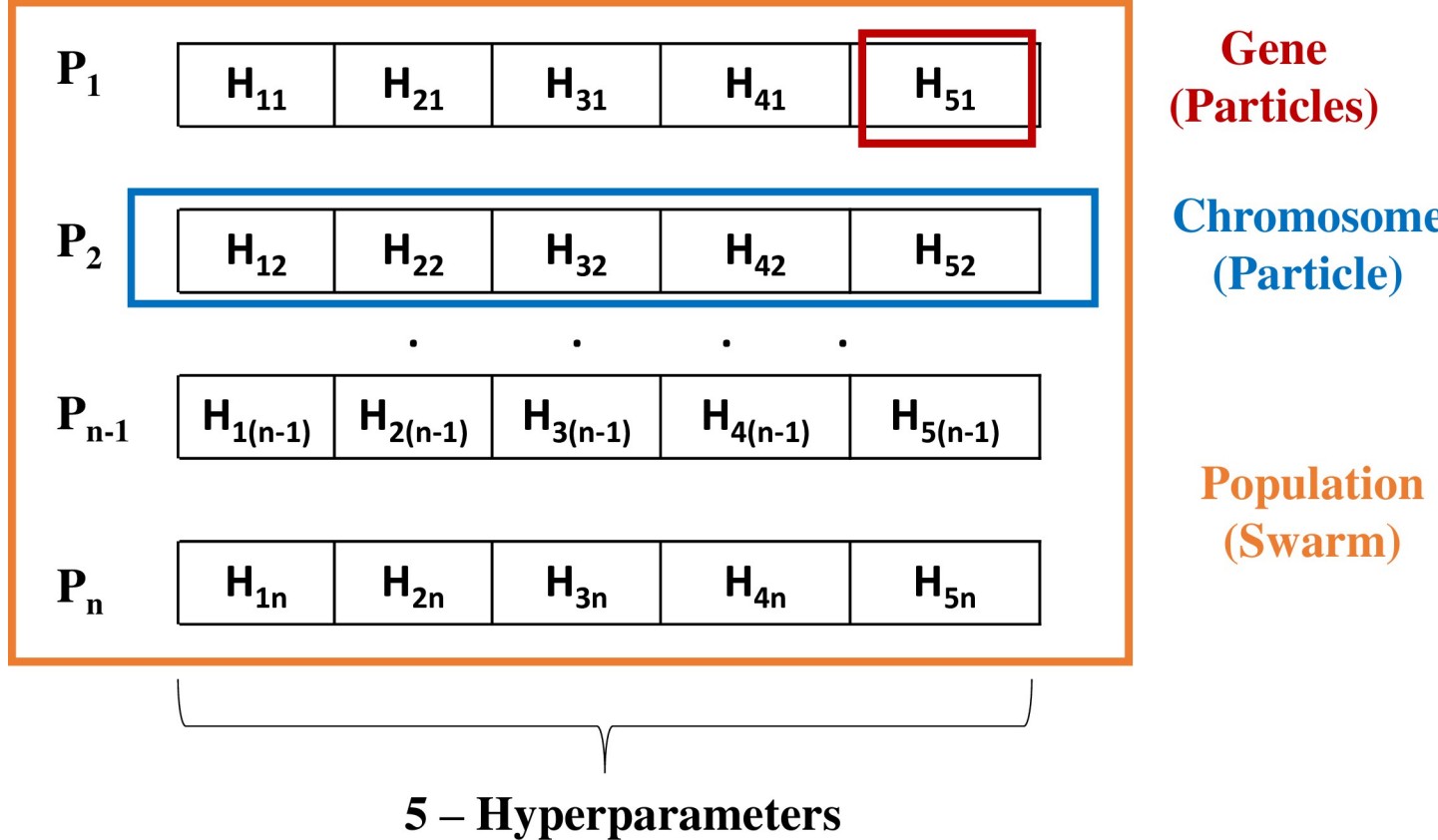

**Fig 6. The architecture of the population (the swarm).**

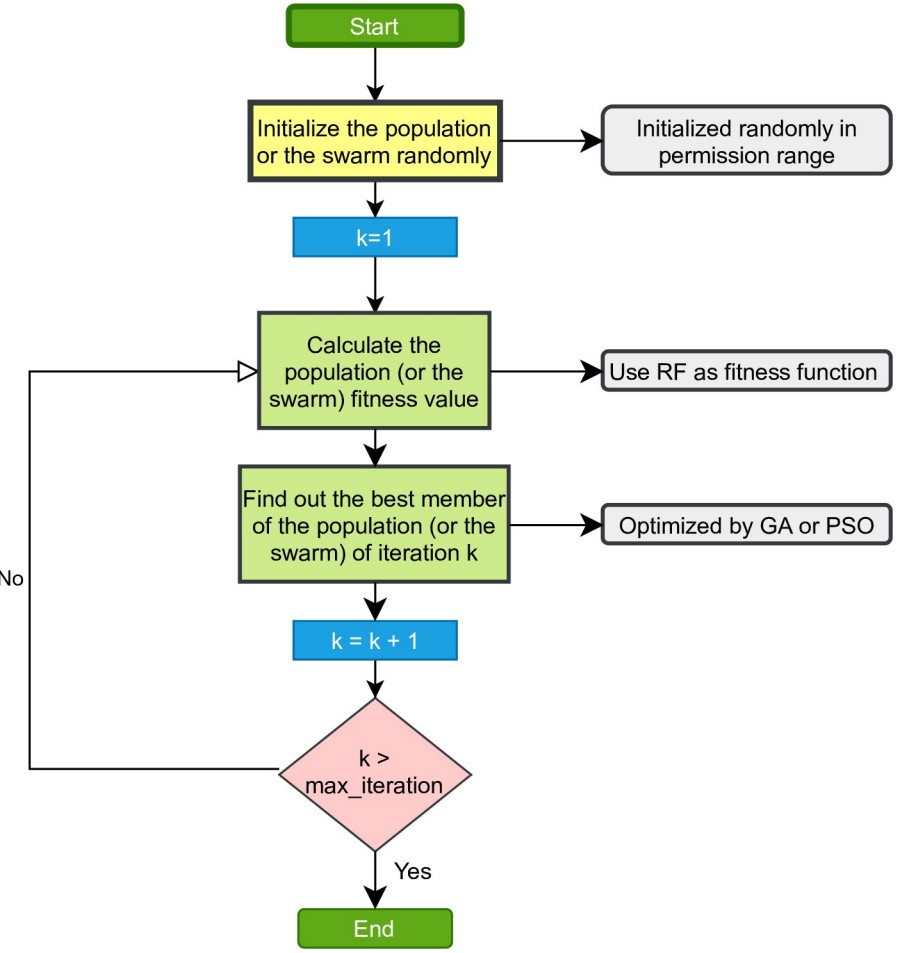

**Fig 7. The flowchart of hybrid RF models.**

## 4.4. Performance evaluation

In this paper, three indicators accounting for the error between the actual and predicted values were used, namely the mean absolute error (MAE), root mean square error (RMSE), squared correlation coefficient ($R^2$). The $R^2$ measures the squared correlation between the predicted and actual values, having values in the range of [0, 1]. Low RMSE and MAE show better accuracy of the proposed ML algorithms. On the other hand, RMSE calculates the squared root average difference, whereas MAE calculates the difference between the predicted and actual values. These values can be calculated using the following equations [58–60]:

$$\text{MAE} = \frac{1}{k} \sum_{i=1}^{k} |v_i - \bar{v}_i| \tag{1}$$

$$\text{RMSE} = \sqrt{\frac{1}{k} \sum_{i=1}^{k} (v_i - \bar{v}_i)^2} \tag{2}$$

**Table 3. The initial values of optimization algorithms.**

| RF-GA | | RF-PSO | |
|---|---|---|---|
| Parameter | Value | Parameter | Value |
| Population | 30 | Number of particles | 30 |
| Number of children | 12 | C1 | 1.4 |
| Mutation rate | 0.4 | C2 | 2 |
| Generation | 100 | w | 1 |
| Fitness value | $R^2$ | $w_d$ | 0.99 |
| Data | Training set/10-Fold CV | Fitness value | $R^2$ |
| | | Data | Training set/10-Fold CV |
| | | Iteration | 100 |

$$R^2 = 1 - \frac{\sum_{i=1}^{k} \left(v_i - \bar{v}_i\right)^2}{\sum_{i=1}^{k} \left(v_i - \bar{v}\right)^2} \tag{3}$$

where $k$ infers the number of the samples, $v_i$, and $\bar{v}_i$ are the actual and predicted outputs, respectively, and $\bar{v}$ is the average value of the $v_i$.

## 5. Results and discussion

### 5.1. Hyperparameters tuning

Fig 8 showed the performance of the RF-GA and RF-PSO models after 20 runs with the random initiation population. It can be seen that the performance of the models after each run is

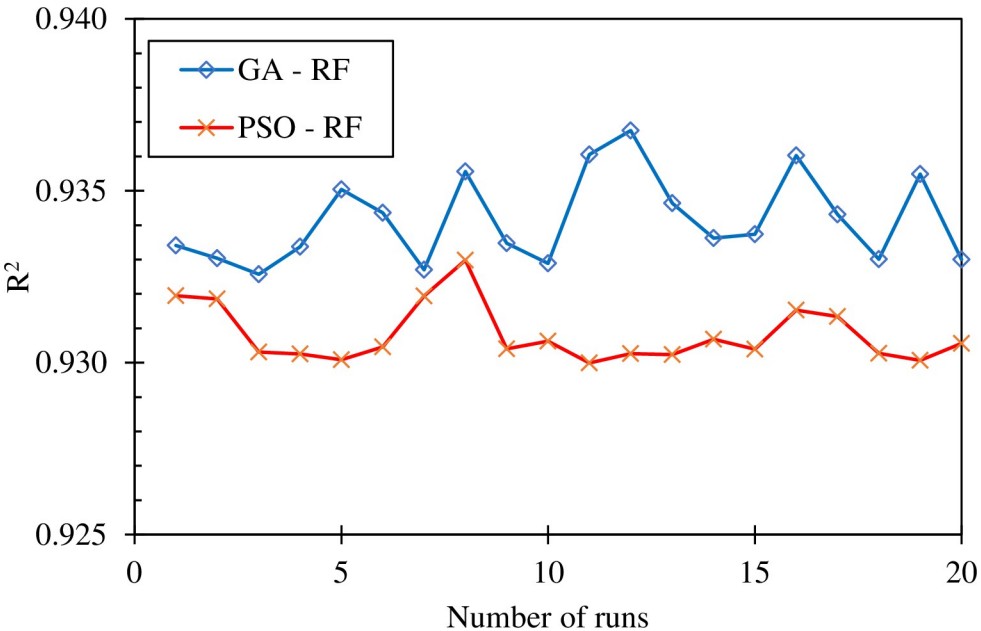

**Fig 8. Best value of $R^2$ in 20 optimization runs using RF-GA and RF-PSO.**

different. Specifically, the best result of the RF-GA R2 model is 0.937 in the 12th round while the RF-PSO model gives the best result with R2 reaching 0.933 in the 8th round. In addition, the RF-GA model gives the lowest result was R2 = 0.932 at the 3rd iteration, while the RF-PSO model achieved the worst result R2 = 0.929 at the 11th iteration. Overall, the RF-GA model seemed to be more efficient compared with the RF-PSO model.

Fig 9 illustrated the best results of the RF-GA and RF-PSO models after 100 iterations. It can be seen that the RF-GA model converged quickly and achieved the best results after the 34th iteration with a performance index R2 = 0.937. On the other side, the RF-PSO model appeared to be slower in convergence and had the best result of only R2 = 0.933 at the 94th loop. The loop increase may continue to give better performance for both two-hybrid models, however, in the framework of this study, 100 loops is the limit to compare the performance of two hybrid models.

The best hyper-parameters combinations found through the RF-GA and RF-PSO models were given in Table 4. It is worth noting that the min_sample_leaf value of the two optimal models is equal to 2, the other hyper-parameters were not the same.

## 5.2. Performance comparison of RF, RF-PSO, RF-GA

From a statistical probability standpoint, the randomness of the division of training and test datasets should be carefully considered. In this step, 1000 random samplings of the training set and testing set were performed to verify the stability of the models. Specifically, three models were compared including RF-GA, RF-PSO, and RF, where the RF model is used with default parameters.

Fig 10 showed the density graph of the models after 1000 runs for the performance indicators such as R2, RMSE, and MAE for training and testing part of 3 models while the results were summarized in Tables 5–7 for $R^2$, RMSE, and MAE, respectively. The results showed that, on the training set, the RF-PSO model gives the best results with the average performance indicators reaching $R^2$ = 0.982, RMSE = 0.04649 and MAE = 0.033, respectively. However, on the testing set, the RF-PSO model gave bad results when the performance indicators were $R^2$ =

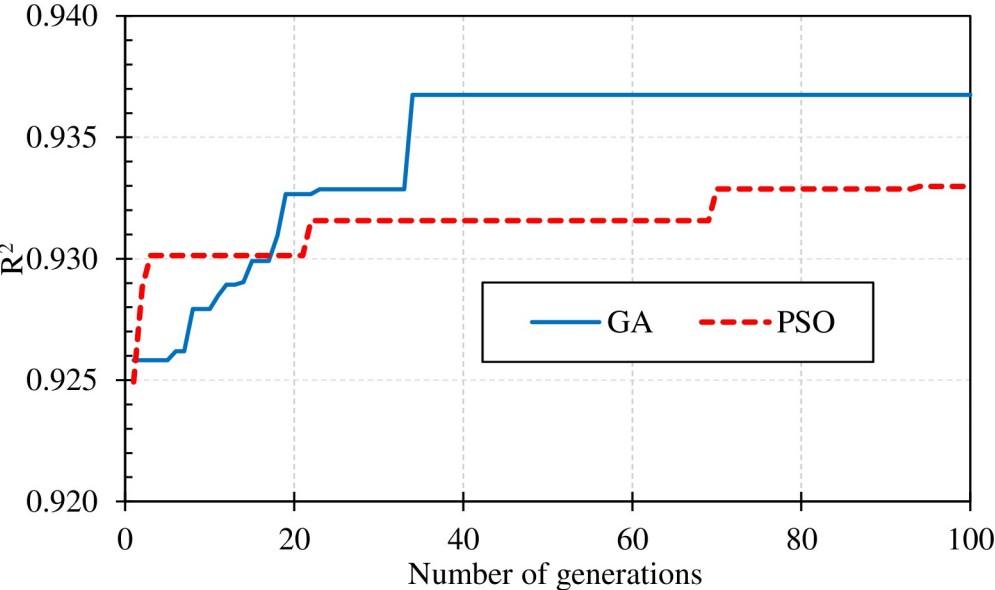

**Fig 9. Hyper-parameters tuning using RF-GA and PSO-GA.**

**Table 4. Best parameters proposed by GA and PSO algorithms.**

|        | Max_depth | Max_features | Min_samples_leaf | Min_samples_split | n_estimators |
|--------|-----------|--------------|------------------|-------------------|--------------|
| RF-GA  | 12        | 1            | 2                | 12                | 144          |
| RF-PSO | 8         | 2            | 2                | 6                 | 16           |

0.9242, RMSE = 0.0939, and MAE = 0.0679, respectively. This result is equivalent to the default-RF model when achieving the corresponding indicators $R^2$ = 0.9240, RMSE = 0.0940, and MAE = 0.0680.

This implies that the RF-PSO and default-RF models proved too fit for the training set and not good in the testing set. In other words, these models are a bit overfitting and do not generalize well the data set. In the opposite direction, the RF-GA model showed better generality when the results were good on the test set with the best average performance indexes and achieved $R^2$ = 0.93043, RMSE = 0, 08847, MAE = 0.06603, and the corresponding results achieved on the training set are $R^2$ = 0.96311, RMSE = 0.06545, MAE = 0.04953 respectively. In addition, the standard deviation of the RF-GA model on the test set is also the smallest in all 3 criteria, proving that the model has the best stability. Generally, in terms of stability and best generalization, the RF-GA model was selected as the last model in this study.

## 5.3. Prediction performance of hybrid model RF-GA

The best architecture of the RF model determined by GA algorithms was applied for this section. In this section, the predictive capacity of the best-performance RF-GA model was presented. In especially, the best RF architecture's prediction results were presented.

A regression model in Fig 11 showed the correlation between the actual and predicted values for the training and testing datasets, respectively. A linear fit was also applied and plotted in each case. It is observed that the linear regression lines were very close to the diagonal lines, which confirms the close correlation between the actual and predicted axial bearing capacity of piles. The calculated values of $R^2$, RMSE and MAE for the training dataset were 0.9639, 0.0661, 0.0511 and 0.9331, 0.0929, 0.0675 for the testing dataset, respectively. The results of the performance criteria show that the RF model with the tuned hyperparameters can accurately predict the axial bearing capacity of piles. Fig 12 showed the error values corresponding to the training and testing databases are low. Almost all the error values between the actual and predicted values were about 0 for the training and testing part confirmed that the RF model has been successful in estimating the axial load capacity of the pile.

It is important to note that due to the limitations of this study, the best RF model developed only achieves high prediction performance under the condition that the input parameter values are between the minimum and maximum values. Input values that are outside the recommended range will cause the model to be confused and incorrectly predict the bearing capacity of the piles.

Moreover, the range of input and output values is crucial in improving performance of ML model [22]. In this investigation, the value of axial bearing capacity of pile varies about from 0.5 to 1.5 MN. However, the missing value of range (0.7;1.0) for the axial bearing capacity of pile (cf. Fig 11) seems to reduce the performance of RF-GA model. The prediction of axial bearing capacity of pile in this missing range needs to be careful and is not recommended. With the database containing 472 samples and 10 input variables, the prediction of axial bearing capacity by RF-GA model is recommended in range (0.40;0.70) and (1.00;1.55) MN of axial bearing capacity of pile. Therefore, the performance and reliability of prediction can be improved if the missing of range is completed in future research.

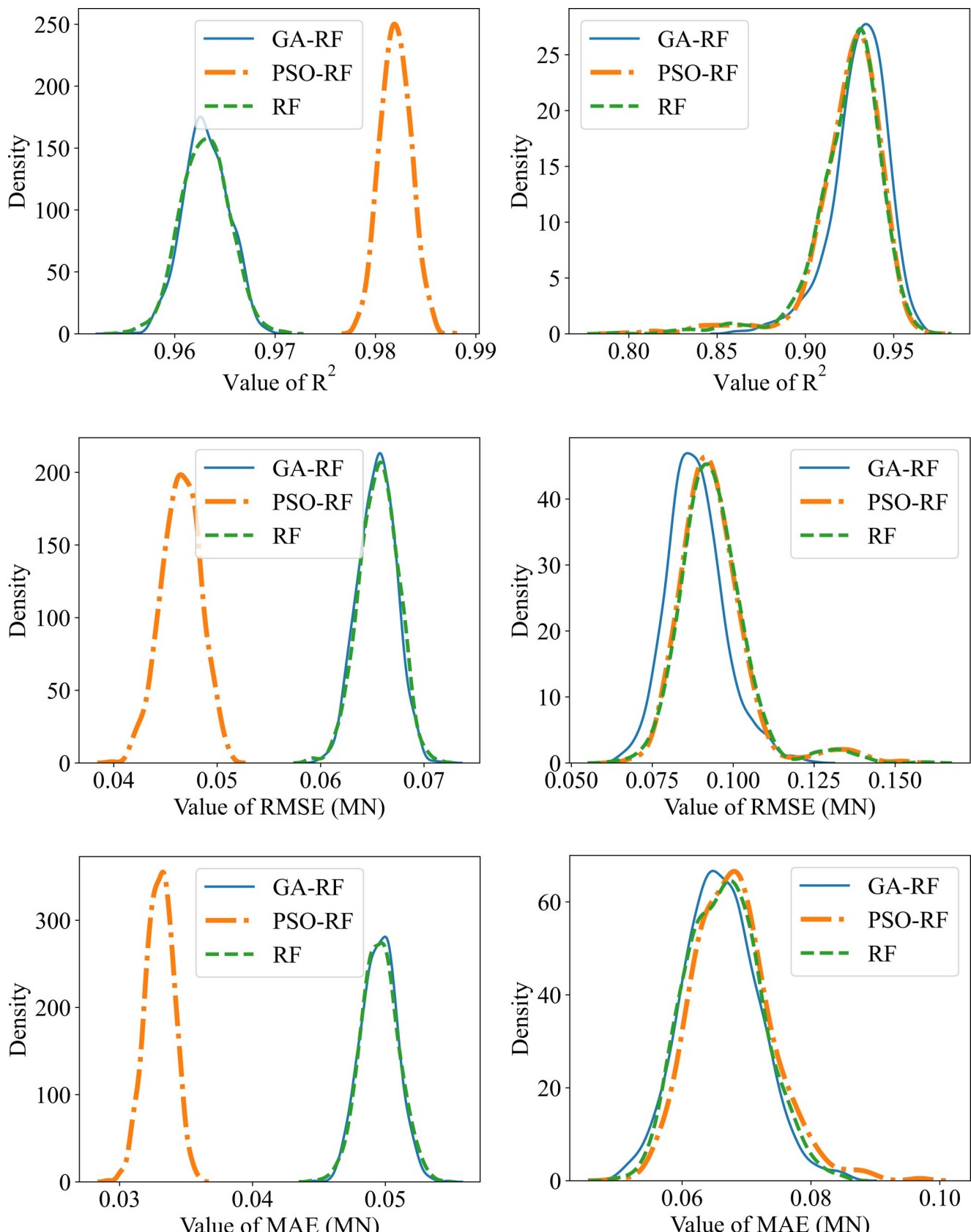

**Fig 10.** Density chart of models after 1000 runs for data set: (a) (c) (e)–training set; (b) (d) (f)–testing set.

**Table 5. Summary of the 1000 simulations using R2 criteria.**

| Model | Dataset | Average | Min | Max | SD |
|---|---|---|---|---|---|
| RF-GA | Training | 0.96311 | 0.9539 | 0.96947 | 0.00223 |
| | Testing | 0.93043 | 0.8587 | 0.96487 | 0.01574 |
| RF-PSO | Training | 0.98201 | 0.97776 | 0.98739 | 0.0015 |
| | Testing | 0.92417 | 0.79826 | 0.96367 | 0.02054 |
| RF | Training | 0.963 | 0.95474 | 0.97103 | 0.00239 |
| | Testing | 0.92404 | 0.79143 | 0.96765 | 0.01996 |

In practical engineering application, the RF model can be illustrated via a large number of decision trees which is built in the form of if-else structures in EXCEL, so the user only needs to enter 10 input variables and get the output variable, which is the pile load capacity. The EXCEL file of load capacity estimation which contains the final RF model is attached in S1 Data.

### 5.4. Sensitivity analysis

The RF algorithm is capable of evaluating the importance of the input parameters. The importance of each input variable is measured by the change in the accuracy of the prediction when the input variable is not selected during the division for each decision tree [61]. The importance of the variables is represented by the Shapley Additive Explanations [62]. This type of plot aggregates SHAP values for all the features is shown in Fig 13. According to the SHAP value, the X8 input corresponding to pile tip elevation is the most important feature. The pile tip elevation has a positive impact on the axial bearing capacity of the pile, in fact, with higher pile tip elevation, the bearing capacity is increased. However, the X5 input corresponding to natural ground elevation has a negative impact on the bearing capacity of the pile. With the lower natural ground elevation, the bearing capacity is higher. These behavior are concluded by Coyle et Sulaiman [63] and Liu et al. [64]. The lowest impact on the pile bearing capacity is pile top elevation which has a positive effect. With a higher elevation of pile top, the pile bearing capacity is slightly increased.

## 6. Conclusion

In this study, RF hybrid models were developed to predict the axial load capacity of the pile. Two global optimization algorithms, GA and PSO, were selected for the hyperparameters optimization of the RF model. For research purposes, the data set including 472 pile load test results were used to train and test the model. The results show that out of the 2 optimized algorithms selected, GA seems to provide better performance than PSO in optimizing the RF model. Specifically, the RF-GA model gives good results in the training set,

**Table 6. Summary of the 1000 simulations using RMSE criteria.**

| Model | Dataset | Average | Min | Max | SD |
|---|---|---|---|---|---|
| RF-GA | Training | 0.06545 | 0.05966 | 0.07226 | 0.0018 |
| | Testing | 0.08847 | 0.06472 | 0.12442 | 0.00889 |
| RF-PSO | Training | 0.04649 | 0.03982 | 0.05132 | 0.00188 |
| | Testing | 0.09391 | 0.06768 | 0.15244 | 0.01117 |
| RF | Training | 0.0656 | 0.05878 | 0.07148 | 0.00188 |
| | Testing | 0.09403 | 0.06329 | 0.15929 | 0.01097 |

**Table 7. Summary of the 1000 simulations using MAE criteria.**

| Model | Dataset | Average | Min | Max | SD |
|---|---|---|---|---|---|
| RF-GA | Training | 0.04953 | 0.04472 | 0.05476 | 0.00133 |
| | Testing | 0.06603 | 0.05014 | 0.08567 | 0.00582 |
| RF-PSO | Training | 0.033 | 0.02913 | 0.03595 | 0.00105 |
| | Testing | 0.06788 | 0.05007 | 0.09725 | 0.00621 |
| RF | Training | 0.04957 | 0.04454 | 0.05417 | 0.0014 |
| | Testing | 0.06801 | 0.04923 | 0.09708 | 0.00622 |

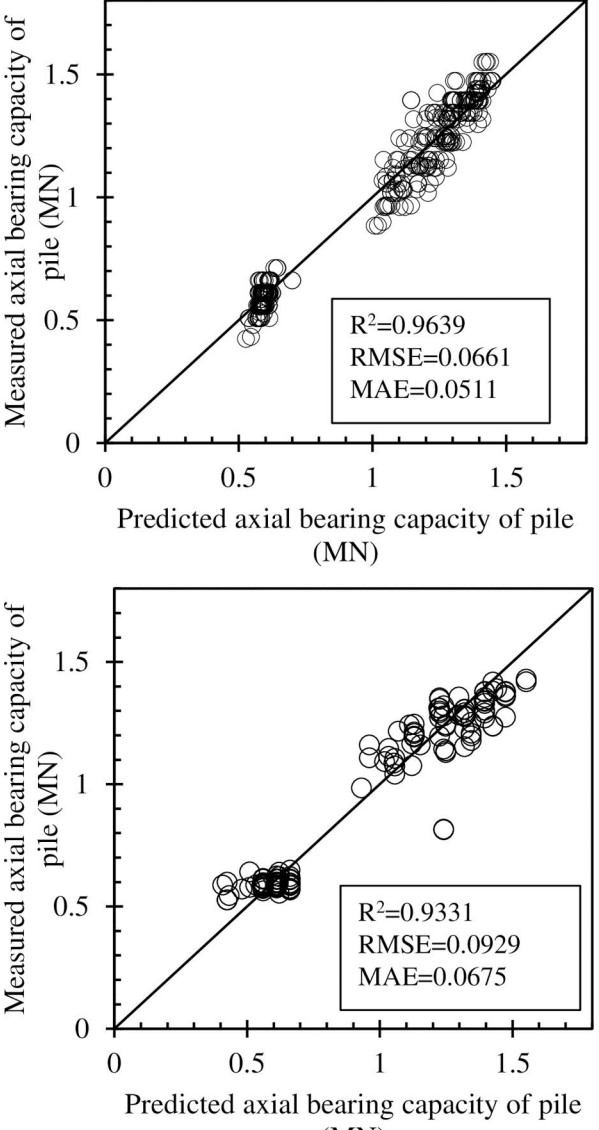

**Fig 11.** Regression graphs for the case of the best parameters of RF-GA model (a) training dataset; and (b) testing dataset.

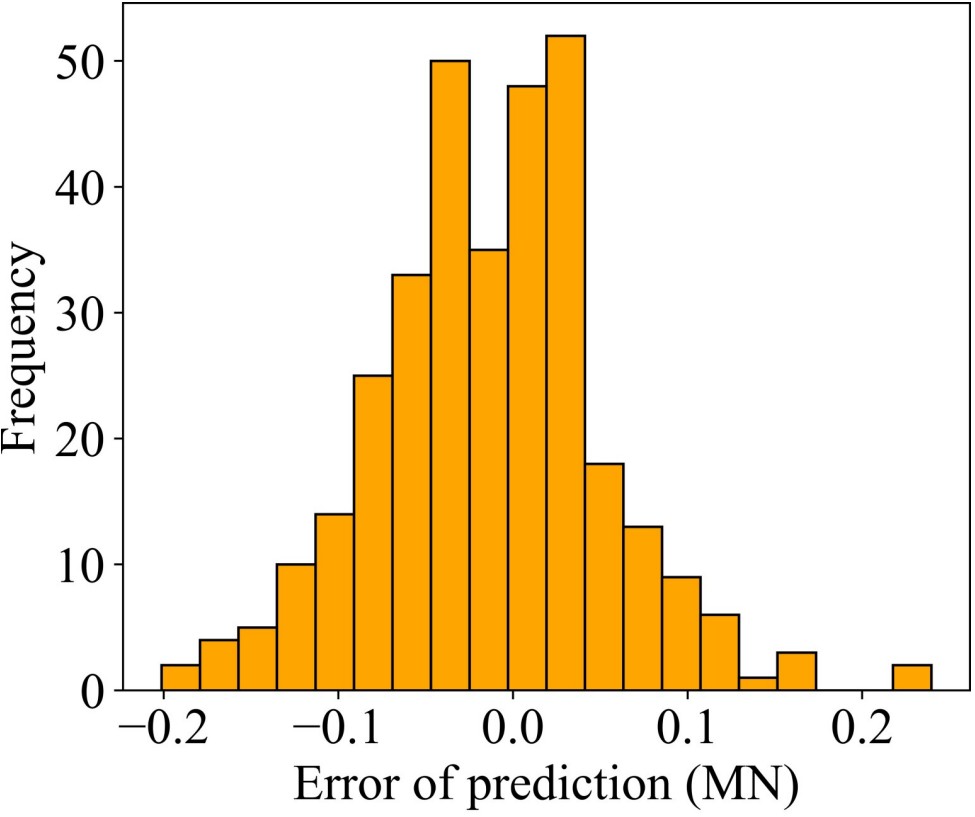

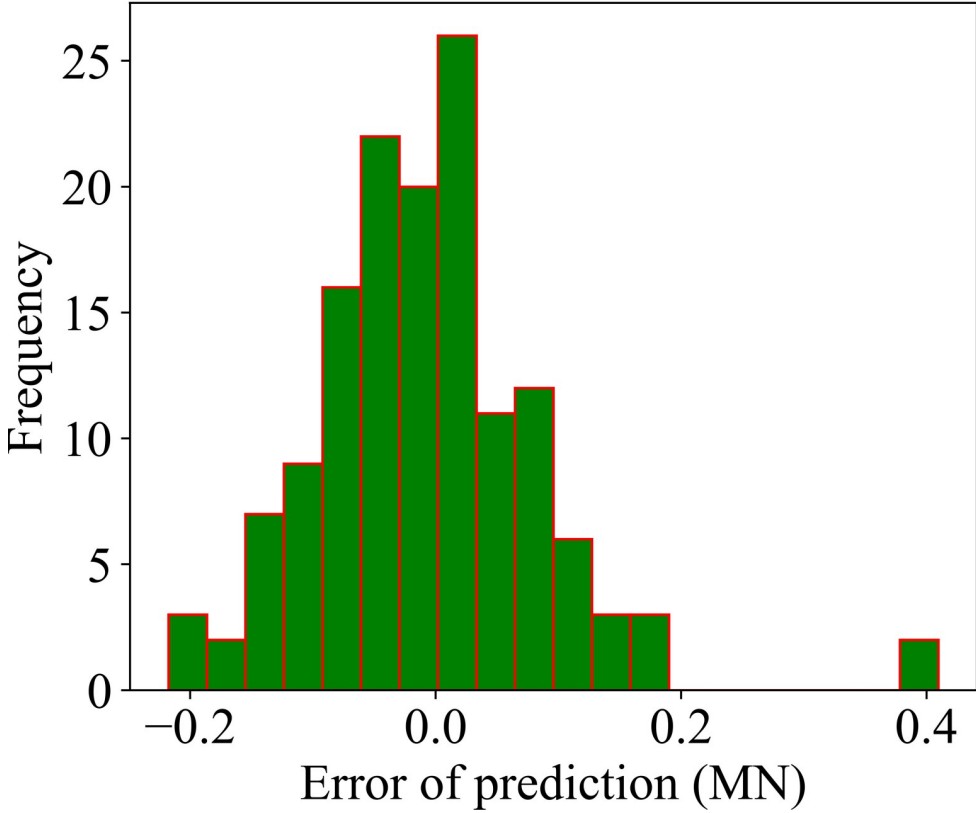

**Fig 12.** Error between target and output values plots for the case of the best model RF-GA (a) training dataset; and (b) testing dataset.

while also providing expert performance on the test set. Meanwhile, the RF-PSO model appears to be overfitting when it comes to excellent performance on the training set, but poorly on the testing set. In addition, when compared to the default RF model, both the RF-GA model and the RF-PSO model yield better results demonstrating the efficiency when using the optimal algorithms. In addition, a sensitivity analysis using the RF-GA model showed that amongst 10 input variables used to predict the axial bearing capacity of the pile, the pile tip elevation was the most important feature, this feature has a positive effect on the axial bearing capacity of piles.

Overall, the RF model optimized by GA provides expert performance in predicting the axial load capacity of the pile. This model could be used as a quick and accurate tool to predict the axial load capacity of the pile. In addition, the model also has great potential in solving other technical problems. In order to increase the performance and reliability of ML model in predicting axial bearing capacity of pile, the range of axial bearing capacity of pile in [0.70;1.00] and the associated input variable values need to be completed.

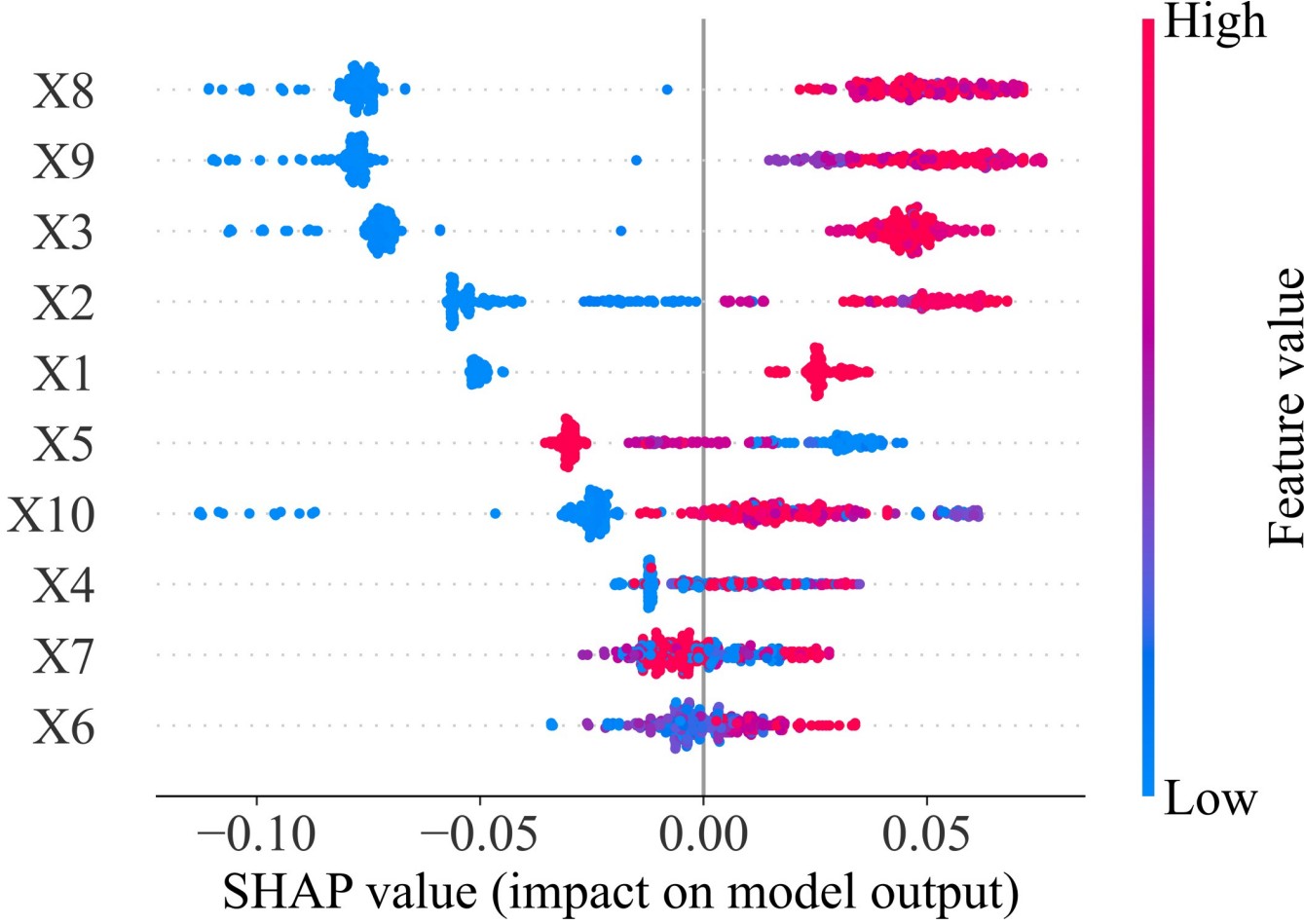

**Fig 13. Feature importance of 10 variables used in this investigation.**

## Supporting information

**S1 Data. Load capacity estimation.**
(XLSX)

## Author Contributions

**Conceptualization:** Tuan Anh Pham, Van Quan Tran.

**Data curation:** Tuan Anh Pham.

**Investigation:** Tuan Anh Pham, Van Quan Tran.

**Methodology:** Van Quan Tran.

**Writing – original draft:** Tuan Anh Pham, Van Quan Tran.

**Writing – review & editing:** Tuan Anh Pham, Van Quan Tran.

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
