## [Decision Letter · Decision Letter 0]

21 Jan 2022

PONE-D-21-34770Developing random forest hybridization models for estimating the axial bearing capacity of pilePLOS ONE

Dear Dr. Tran,

Thank you for submitting your manuscript to PLOS ONE. After careful consideration, we feel that it has merit but does not fully meet PLOS ONE’s publication criteria as it currently stands. Therefore, we invite you to submit a revised version of the manuscript that addresses the points raised during the review process.

We look forward to receiving your revised manuscript.

Kind regards,

Wajid Mumtaz

Academic Editor

PLOS ONE

Journal Requirements:

Reviewers' comments:

Reviewer's Responses to Questions

**Comments to the Author**

1. Is the manuscript technically sound, and do the data support the conclusions?

Reviewer #1: Yes

2. Has the statistical analysis been performed appropriately and rigorously? 

Reviewer #1: Yes

3. Have the authors made all data underlying the findings in their manuscript fully available?

Reviewer #1: Yes

4. Is the manuscript presented in an intelligible fashion and written in standard English?

Reviewer #1: Yes

5. Review Comments to the Author

Reviewer #1: In this article, the authors present soft computing techniques to predict the axial bearing capacity of piles. Especially, artificial intelligent techniques such as Random Forest (RF) models for the prediction of the axial bearing capacity of piles are developed and proposed. The problem of the estimation of the axial bearing capacity of piles is an important significant issue in structural and geotechnical engineering, and such an attempt is of great interest. However the paper, in its present form, requires some substantial modifications in order to justify its publication in an International Journal such as Plos One. The following points should be further elaborated by the authors:

1. Authors are kindly requested in order to further strengthen their work to add/include a new short section after the Introduction titled Research Significance where they should to justify the need for further research on the subject.

2. The literature review is not adequately covered and complete as presented in the manuscript. Extensive and in-depth state-of-the-art reports can be found in the following works: [Armaghani, D.J.; Asteris, P.G.; Fatemi, S.A.; Hasanipanah, M.; Tarinejad, R.; Rashid, A.S.A.; Huynh, V.V. On the Use of Neuro-Swarm System to Forecast the Pile Settlement. Appl. Sci. 2020, 10, 1904. https://doi.org/10.3390/app10061904; Asteris, P.G., Skentou, A.D., Bardhan, A., Samui, P., and Pilakoutas, K. (2021). Predicting Concrete Compressive Strength using Hybrid Ensembling of Surrogate Machine Learning Models, Cement and Concrete Research, Volume 145, 106449; Le, T.-T., Asteris, P.G., Lemonis, M.E. (2020). Axial Load Capacity of Rectangular Concrete-filled Steel Tube Columns using Machnine Learning Techniques, Engineering with Computers, https://doi.org/10.1007/s00366-021-01461-0; Apostolopoulou, M., Asteris, P.G., Armaghani, D.J., Douvika, MG., Lourenço, P.B., Cavaleri, L., Bakolas, A., Moropoulou, A. (2020). Mapping and holistic design of natural hydraulic lime mortars, Cement and Concrete Research, 136, 106167, https://doi.org/10.1016/j.cemconres.2020.106167; Armaghani, D.J., Mamou, A., Maraveas, C., Roussis, P.C., Siorikis, V.G., Skentou, A.D., Asteris, P.G. (2021). Predicting the unconfined compressive strength of granite using only two non-destructive test indexes, Geomechanics and Engineering, 317-330]. An additional paragraph, containing the augmented literature review, should be added.

3. The authors are kindly requested to include a short paragraph about the limitations of their work. Namely, they should to make clear that their proposed models are valid for input parameters values among the minimum and maximum values of the ten input parameters.

4. Also, authors should include a comment about the reliability of the database used. As a general trend, it is noticed that, during the process of developing a forecast model, researchers pay particular attention to the computational model itself, while at the same time, not giving the same amount of attention to the database that is used for the development, training and validation of the model. Although research related to new computational models is of course of high importance and added value for the international scientific community, the authors believe that, since the ultimate goal is a reliable forecast, the reliability of the database should be of utmost importance and should be thoroughly investigated in this regard. In fact, a reliable database must comprise of not only reliable data, but also of a sufficient amount of data, that covers the full range of parameter values, regarding the parameters which influence the problem investigated [Asteris, P.G., Apostolopoulou, M., Armaghani, D.J., Cavaleri, L., Chountalas, A.T., Guney, D., Hajihassani, M., Hasanipanah, M., Khandelwal, M., Karamani, C., Koopialipoor, M., Kotsonis, E., Le, T-T., Lourenço, P.B., Ly, H-B., Moropoulou, A., Nguyen, H., Pham, B.T., Samui, P., Zhou, J. (2020). On the metaheuristic models for the prediction of cement-metakaolin mortars compressive strength, Metaheuristic Computing and Applications, 1(1), 63-99, DOI: http://dx.doi.org/10.12989/mca.2020.1.1.063]. Based on the above their database which consists of only 99 datasets while at the same time their input parameters are 10 they should to include a comment about this issu.

5. It is well known the majority of authors present in their published articles only the architecture of NN model. Any architecture without the values of final values of NN model weights has very little value for others researchers and practicing engineers. In order to be useful, a proposed NN architecture should be accompanied by the (quantitative) values of weights. Authors are kindly requested to present their models final values of weights and bias [Asteris, P.G., Mokos, V.G. (2020). Concrete Compressive Strength using Artificial Neural Networks, Neural Computing and Applications, 32, 1807–11826, https://doi.org/10.1007/s00521-019-04663-2 ; Armaghani, D.J., Asteris, P.G. (2020). A comparative study of ANN and ANFIS models for the prediction of cement-based mortar materials compressive strength, Neural Computing and Applications, https://doi.org/10.1007/s00521-020-05244-4 ; Duan, J., Asteris, P.G., Nguyen, H. Bui, X.-N., Moayedi, H. (2020). A Novel Artificial Intelligence Technique to Predict Compressive Strength of Recycled Aggregate Concrete Using ICA-XGBoost Model, Engineering With Computers, https://doi.org/10.1007/s00366-020-01003-0; Zeng, J., Roussis, P.C., Mohammed, A.S., Maraveas C.,Fatemi S.A., Armaghani, D.J., Asteris, P.G. (2021).Prediction of peak particle velocity caused by blasting through the combinations of boosted-chaid and svm models with various kernels, Applied Sciences (Switzerland), 2021, 11(8), 3705; ].

6. PLOS authors have the option to publish the peer review history of their article (what does this mean?). If published, this will include your full peer review and any attached files.

Reviewer #1: **Yes: **Panagiotis G. Asteris

---

## [Author Response · Author response to Decision Letter 0]

16 Feb 2022

RESPONSES OF THE ACADEMIC EDITOR AND REVIEWERS’ COMMENTS

I. RESPONSE TO REVIEWER #1

Comment #1. Authors are kindly requested in order to further strengthen their work to add/include a new short section after the Introduction titled Research Significance where they should to justify the need for further research on the subject.

Response:

We thank Reviewer #1 for this recommendation. A new section “Research Significance” is added in the revised manuscript.

Comment #2. The literature review is not adequately covered and complete as presented in the manuscript. Extensive and in-depth state-of-the-art reports can be found in the following works: 

Armaghani, D.J.; Asteris, P.G.; Fatemi, S.A.; Hasanipanah, M.; Tarinejad, R.; Rashid, A.S.A.; Huynh, V.V. On the Use of Neuro-Swarm System to Forecast the Pile Settlement. Appl. Sci. 2020, 10, 1904. https://doi.org/10.3390/app10061904;

Asteris, P.G., Skentou, A.D., Bardhan, A., Samui, P., and Pilakoutas, K. (2021). Predicting Concrete Compressive Strength using Hybrid Ensembling of Surrogate Machine Learning Models, Cement and Concrete Research, Volume 145, 106449; 

Le, T.-T., Asteris, P.G., Lemonis, M.E. (2020). Axial Load Capacity of Rectangular Concrete-filled Steel Tube Columns using Machnine Learning Techniques, Engineering with Computers, https://doi.org/10.1007/s00366-021-01461-0;

Apostolopoulou, M., Asteris, P.G., Armaghani, D.J., Douvika, MG., Lourenço, P.B., Cavaleri, L., Bakolas, A., Moropoulou, A. (2020). Mapping and holistic design of natural hydraulic lime mortars, Cement and Concrete Research, 136, 106167, https://doi.org/10.1016/j.cemconres.2020.106167;

Armaghani, D.J., Mamou, A., Maraveas, C., Roussis, P.C., Siorikis, V.G., Skentou, A.D., Asteris, P.G. (2021). Predicting the unconfined compressive strength of granite using only two non-destructive test indexes, Geomechanics and Engineering, 317-330]. 

An additional paragraph, containing the augmented literature review, should be added.

Response:

We thank Reviewer #1 for this recommendation. A new paragraph containing the augmented literature review is added in the revised manuscript.

Comment #3. The authors are kindly requested to include a short paragraph about the limitations of their work. Namely, they should to make clear that their proposed models are valid for input parameters values among the minimum and maximum values of the ten input parameters.

Response:

We appreciate reviewer comments. We appreciate reviewer comments. This comment is completely worthwhile because, by nature, decision tree-based models like Random Forest will not be able to predict outside of the trained range. Therefore, we added the following limitation of the study in the resubmission:

“It is important to note that due to the limitations of this study, the best RF model developed only achieves high prediction performance under the condition that the input parameter values are between the minimum and maximum values. Input values that are outside the recommended range will cause the model to be confused and incorrectly predict the bearing capacity of the piles.”

Comment #4. Also, authors should include a comment about the reliability of the database used. As a general trend, it is noticed that, during the process of developing a forecast model, researchers pay particular attention to the computational model itself, while at the same time, not giving the same amount of attention to the database that is used for the development, training and validation of the model. Although research related to new computational models is of course of high importance and added value for the international scientific community, the authors believe that, since the ultimate goal is a reliable forecast, the reliability of the database should be of utmost importance and should be thoroughly investigated in this regard. In fact, a reliable database must comprise of not only reliable data, but also of a sufficient amount of data, that covers the full range of parameter values, regarding the parameters which influence the problem investigated 

Asteris, P.G., Apostolopoulou, M., Armaghani, D.J., Cavaleri, L., Chountalas, A.T., Guney, D., Hajihassani, M., Hasanipanah, M., Khandelwal, M., Karamani, C., Koopialipoor, M., Kotsonis, E., Le, T-T., Lourenço, P.B., Ly, H-B., Moropoulou, A., Nguyen, H., Pham, B.T., Samui, P., Zhou, J. (2020). On the metaheuristic models for the prediction of cement-metakaolin mortars compressive strength, Metaheuristic Computing and Applications, 1(1), 63-99, 

DOI: http://dx.doi.org/10.12989/mca.2020.1.1.063.

Based on the above their database which consists of only 99 datasets while at the same time their input parameters are 10 they should to include a comment about this issue.

Response:

We thank Reviewer #1 for this excellent recommendation and discussion about reliability of database. Based on the interesting comment of Reviewer #1, a new paragraph is added in section 5.3 and conclusion of the revised manuscript. 

Comment #5. It is well known the majority of authors present in their published articles only the architecture of NN model. Any architecture without the values of final values of NN model weights has very little value for others researchers and practicing engineers. In order to be useful, a proposed NN architecture should be accompanied by the (quantitative) values of weights. Authors are kindly requested to present their models final values of weights and bias 

Asteris, P.G., Mokos, V.G. (2020). Concrete Compressive Strength using Artificial Neural Networks, Neural Computing and Applications, 32, 1807–11826, https://doi.org/10.1007/s00521-019-04663-2 ; Armaghani, D.J., Asteris, P.G. (2020). A comparative study of ANN and ANFIS models for the prediction of cement-based mortar materials compressive strength, Neural Computing and Applications, https://doi.org/10.1007/s00521-020-05244-4 ;

Duan, J., Asteris, P.G., Nguyen, H. Bui, X.-N., Moayedi, H. (2020). A Novel Artificial Intelligence Technique to Predict Compressive Strength of Recycled Aggregate Concrete Using ICA-XGBoost Model, Engineering With Computers, https://doi.org/10.1007/s00366-020-01003-0;

Zeng, J., Roussis, P.C., Mohammed, A.S., Maraveas C.,Fatemi S.A., Armaghani, D.J., Asteris, P.G. (2021). Prediction of peak particle velocity caused by blasting through the combinations of boosted-chaid and svm models with various kernels, Applied Sciences (Switzerland), 2021, 11(8), 3705;

Response:

We appreciate the reviewer's comment. We agree that in practice, engineers and researchers will find it difficult to apply studies if only information about the architecture of the models is available. However, due to the nature of the Random Forest model containing a large number of decision trees, a complete illustration of decision trees is difficult within the framework of the manuscript. Therefore, we suggest moving the final RF model to an open spreadsheet (specifically, the one in MS EXCEL). The decision trees model is built in the form of if-else structures in EXCEL, so the user only needs to enter 10 input variables and get the output variable, which is the pile load capacity. The EXCEL file which contains the final RF model is attached with the revised manuscript.

---

## [Decision Letter · Decision Letter 1]

8 Mar 2022

Developing random forest hybridization models for estimating the axial bearing capacity of pile

PONE-D-21-34770R1

Dear Dr. Van Quan Tran,

We’re pleased to inform you that your manuscript has been judged scientifically suitable for publication and will be formally accepted for publication once it meets all outstanding technical requirements.

Kind regards,

Wajid Mumtaz

Academic Editor

PLOS ONE

Additional Editor Comments (optional):

Reviewers' comments:

Reviewer's Responses to Questions

**Comments to the Author**

1. If the authors have adequately addressed your comments raised in a previous round of review and you feel that this manuscript is now acceptable for publication, you may indicate that here to bypass the “Comments to the Author” section, enter your conflict of interest statement in the “Confidential to Editor” section, and submit your "Accept" recommendation.

Reviewer #1: All comments have been addressed

2. Is the manuscript technically sound, and do the data support the conclusions?

Reviewer #1: Yes

3. Has the statistical analysis been performed appropriately and rigorously? 

Reviewer #1: Yes

4. Have the authors made all data underlying the findings in their manuscript fully available?

Reviewer #1: Yes

5. Is the manuscript presented in an intelligible fashion and written in standard English?

Reviewer #1: Yes

6. Review Comments to the Author

Reviewer #1: No The authors have adequately addressed the comments of the reviewers and the manuscript is ready to be sent to the production editor for a final check to see that everything is in order.

7. PLOS authors have the option to publish the peer review history of their article (what does this mean?). If published, this will include your full peer review and any attached files.

Reviewer #1: No

---

## [Editor Report · Acceptance letter]

10 Mar 2022

PONE-D-21-34770R1 

Developing random forest hybridization models for estimating the axial bearing capacity of pile 

Dear Dr. Tran:

I'm pleased to inform you that your manuscript has been deemed suitable for publication in PLOS ONE. Congratulations! Your manuscript is now with our production department. 

Kind regards, 

on behalf of

Dr. Wajid Mumtaz 

Academic Editor

PLOS ONE